# Clinical Experience with Inosine Pranobex in Pediatric Acute Respiratory Infections with Comorbidities: A Case Series from a Specialised Centre

**DOI:** 10.3390/pediatric17060123

**Published:** 2025-11-10

**Authors:** Peter Kunč, Jaroslav Fábry, Katarína Ištvánková, Renata Péčová, Miloš Jeseňák

**Affiliations:** 1Clinic of Paediatric Respiratory Diseases and Tuberculosis, National Institute of Paediatric Tuberculosis and Respiratory Diseases in Dolny Smokovec, Jessenius Faculty of Medicine in Martin, Comenius University in Bratislava, 036 01 Martin, Slovakia; jaroslav.fabry@nudtarch.sk (J.F.); katarina.istvankova@nudtarch.sk (K.I.); 2Department of Pathological Physiology, Jessenius Faculty of Medicine in Martin, Comenius University in Bratislava, 036 01 Martin, Slovakia; renata.pecova@uniba.sk; 3Department of Paediatrics and Adolescent Medicine, Jessenius Faculty of Medicine in Martin, Comenius University in Bratislava, University Hospital in Martin, 036 01 Martin, Slovakia; jesenak@gmail.com; 4Institute of Clinical Immunology and Medical Genetics, Jessenius Faculty of Medicine in Martin, Comenius University in Bratislava, University Hospital Martin, 036 01 Martin, Slovakia

**Keywords:** case series, immunomodulation, inosine pranobex, paediatric acute respiratory infections

## Abstract

**Background:** Acute respiratory infections (ARIs) pose a significant clinical challenge in paediatric populations, especially in children with comorbidities who may exhibit underlying immune dysregulation. Inosine pranobex (IP) is an immunomodulatory agent that enhances T-lymphocyte and Natural Killer (NK) cell function, offering a targeted therapeutic rationale for such cases. **Objective:** This study aimed to retrospectively describe the clinical characteristics, immunological profiles, and outcomes of paediatric patients with complex, PCR-confirmed viral ARIs and significant comorbidities, for whom adjunctive therapy with IP was initiated based on clinical judgment. **Methods:** This retrospective case series analysed data from 14 paediatric patients hospitalised at a specialised centre (National Institute of Paediatric Tuberculosis and Respiratory Diseases in Dolny Smokovec, Slovakia). Cases were selected based on PCR-confirmed viral ARI, a history of recurrent infections, significant comorbidities, and initiation of IP therapy. The indication for IP was guided by the treating physician in cases of severe, prolonged, or recurrent disease course, where immune dysregulation was suspected, often supported by prior immunophenotyping. **Results:** A frequent observation in this cohort was the presence of baseline cellular immune alterations with a frequent observation of baseline cellular immune alterations, most notably the depletion of natural killer (NK) cells. NK cell depletion was identified in half of the patients (7/14). Following the initiation of treatment regimens that included adjunctive IP, clinical stabilisation or improvement was observed in all 14 patients included in the study. The therapy was well tolerated, with no reported adverse events attributable to IP. **Conclusions:** This case series highlights the common presence of cellular immune alterations in children with complex ARIs. While the observational nature of this study precludes any conclusions about causality, the favourable clinical course, safety profile, and strong immunological rationale support the need for prospective controlled trials to evaluate the role of IP in this specific high-risk paediatric population.

## 1. Introduction

Acute respiratory infections (ARIs) represent a significant challenge to healthcare systems globally, contributing substantially to morbidity, particularly in the paediatric population [1]. The aetiology of these infections is predominantly viral, with estimates suggesting that viruses are responsible for over 90% of ARI cases [2]. Epidemiological data consistently indicate that young children, specifically those under the age of five years, experience the highest incidence rates of ARIs, making this demographic a key focus for clinical management and preventive strategies [3,4].

Viral respiratory pathogens damage the airway epithelium, initiating an immune cascade essential for viral clearance. However, this response, which involves the production of interferons and proinflammatory cytokines, can become dysregulated, leading to uncontrolled inflammation and exacerbating pulmonary damage [5,6]. In vulnerable children, particularly those with comorbidities or recurrent infections, this may manifest as deficits in cellular immunity, such as impaired Natural Killer (NK) cell function or T-lymphocyte exhaustion, contributing to a more severe or prolonged disease course [7].

The complex interplay between viral pathogenesis, host immunity, microbial dysbiosis, and the potential for long-term respiratory sequelae underscores the importance of effective management strategies for viral ARIs in children. In this context, therapies that can modulate the host’s immune response rather than solely targeting the pathogen may offer significant benefits [8]. Inosine pranobex (IP) is a synthetic immunomodulatory agent with a well-documented mechanism of action, including the potentiation of a T helper 1 (Th1) response and, crucially, the enhancement of NK cell and cytotoxic T lymphocyte function [9]. This specific mechanism provides a strong therapeutic rationale for its use in patients in whom cellular immune dysfunction is a suspected contributor to the clinical presentation. The purpose of this study was not to compare the efficacy of IP with other immunomodulatory agents, such as bacterial lysates or beta-glucans, but to describe its use in a real-world clinical setting, where it was specifically selected by the attending physician as a targeted intervention. Therefore, this article aimed to retrospectively analyse and describe the clinical and immunological features of a series of complex paediatric cases in which IP was used as adjunctive therapy for acute viral respiratory infections.

## 2. Materials and Methods

This retrospective longitudinal study was conducted at the National Institute of Paediatric Tuberculosis and Respiratory Diseases in Dolny Smokovec, Slovakia, utilising data collected from the hospital information system for paediatric patients hospitalised between January 2022 and May 2025 (total *N* = 1150). This study was approved by the Ethics Committee of the National Institute of Paediatric Tuberculosis and Respiratory Diseases (ID082025).

The patient selection process followed a multi-step approach. Of the total 1150 hospitalized patients during the study period, 346 acquired an ARI during their stay. Among them, a viral aetiology was confirmed in 141 patients using a multiplex real-time PCR (RT-PCR) analysis of laryngeal swabs (AmpliSens^®^ ARVI-screen-FRT PCR kit, Interlabservice, Brno, Czech Republic). This commercial panel screens for a wide range of common respiratory viruses. Cellular immunity parameters, specifically immunophenotyping of lymphocyte subsets performed prior to or at the onset of ARI (in an asymptomatic stage), were available for 32 patients. From this subgroup, the final cohort of 14 patients for whom IP treatment was indicated and who did not meet the exclusion criteria were selected.

The primary inclusion criterion for case selection was hospitalisation for an ARI with a laboratory-confirmed viral aetiology, determined by polymerase chain reaction (PCR) analysis of respiratory specimens, such as oropharyngeal swabs or bronchoalveolar lavage fluid. Additionally, we focused on children with a history of recurrent respiratory infections, defined according to De Martino et al. as meeting at least one of the following conditions: experiencing six or more respiratory infections annually, having at least one upper respiratory tract infection monthly from September to April, or having at least three lower respiratory tract infections annually [10]. The initiation of IP treatment was a crucial inclusion criterion. This decision was made by the attending physician based on clinical judgement in cases of severe, prolonged, or recurrent ARI in patients with significant comorbidities, where insufficient or dysregulated immune response was suspected. For the purpose of this study, significant comorbidities were defined as chronic or recurrent conditions that could potentially influence the patient’s immune response or the clinical course of an ARI. Examples from our cohort include severe persistent asthma, bronchiectasis, gastroesophageal reflux disease (GERD), primary immunodeficiencies, and sequelae of prematurity or congenital malformations. This clinical suspicion was often supported by the results of cellular immunity assessments.

From the initial pool, 32 patients met the inclusion criteria. Participants were subsequently excluded if they had any of the following conditions documented in their medical records: a diagnosis of autoimmune disease, active malignancy, severe hemodynamically significant or uncontrolled cardiovascular disease, major structural congenital defects (e.g., complex cardiac anomalies, diaphragmatic hernia, tracheoesophageal fistula, congenital pulmonary airway malformation), severe uncontrolled primary immunodeficiencies (excluding cases where immunomodulation was part of management for recurrent infections or specific secondary mild deficiencies), clinically relevant malnutrition, cystic fibrosis, diffuse interstitial lung disease unrelated to the acute infectious process, or moderate-to-severe uncontrolled asthma. After applying the exclusion criteria, a final cohort of 14 patients was selected for analysis. The patient selection process is illustrated in Figure 1.

For the final cohort (*N* = 14), data extraction focused on demographics, comorbidities, viral pathogens, and the results of immunological assessments. Cellular immunity was assessed by flow cytometry immunophenotyping of peripheral blood lymphocyte subsets, quantifying total lymphocytes (CD45+), T-lymphocytes (CD3+), T-helper lymphocytes (CD4+), T-suppressor/cytotoxic lymphocytes (CD8+), B-lymphocytes (CD19+), and Natural Killer (NK) cells (CD16+56+). The immunoregulatory CD4/CD8 ratio was also calculated, and the analysis focused particularly on CD8+ T lymphocytes and NK cells because they are known to be critically involved in the antiviral immune response and are also the primary documented targets of IP’s immunomodulatory action. The administration details (dosage, formulation, and duration) of inosine pranobex were also recorded.

## 3. Results

This section presents a case series detailing the clinical course and management of paediatric patients with confirmed ARIs who received inosine pranobex as adjunctive immunomodulatory therapy. The presented cases illustrate the heterogeneity observed in viral aetiologies, underlying patient comorbidities, and immunological profiles at the time of infection, and the comprehensive therapeutic strategies employed.

### 3.1. Case 1

A 17.5-month-old male infant with a history of prematurity and significant familial atopy presented with tracheobronchitis and incipient right-sided bronchopneumonia, initially raising suspicion of foreign body aspiration. PCR analysis of bronchoalveolar lavage (BAL) fluid confirmed co-infection with parainfluenza virus type 3 (PIV3) and Human Bocavirus (HBOV). Concurrent bacterial infection was identified in the BAL (*Staphylococcus aureus*, *Streptococcus haemolyticus* group B, *Haemophilus parainfluenzae*). The initial inflammatory markers were within normal limits. Immunological assessment indicated a marginal reduction in CD19+ B lymphocytes, and other evaluated parameters were within the normal reference ranges without significant deviations. The therapeutic regimen included IP (syrup, 2.5 mL four times daily for 14 days), amoxicillin/clavulanate, inhaled beclometasone dipropionate, inhaled fenoterol/ipratropium bromide, and *Lactobacillus reuteri* probiotics. Clinical improvement led to discharge in a stable condition.

### 3.2. Case 2

A 9-year-old girl with severe persistent asthma, bronchiectasis, allergic rhinitis, and documented therapeutic non-adherence presented with an acute exacerbation manifesting as collapse, dyspnoea, and subsequent interstitial pneumonia. Serological testing indicated a recent infection with respiratory syncytial virus (RSV; IgM positive), which was also confirmed by PCR. Laboratory findings included an elevated erythrocyte sedimentation rate (ESR) with normal C-reactive protein (CRP) and procalcitonin levels. Immunological assessment revealed significant reductions in both the absolute count and percentage of CD8+ T lymphocytes and NK cells. The CD4/CD8 ratio was within the normal range. The treatment comprises IP (250 mg [1/2 tablet] three times daily for approximately 14 days), inhaled fluticasone propionate/salmeterol (ICS/LABA), montelukast (LTRA), and desloratadine. The patient showed clinical improvement.

### 3.3. Case 3

A 7-month-old female infant with chronic bronchitis, gastroesophageal reflux disease (GERD), anaemia, and previous Methicillin-resistant *Staphylococcus aureus* (MRSA) colonisation presented with obstructive bronchitis exacerbation. Nasopharyngeal PCR confirmed PIV3 infection. A throat culture identified bacterial co-infection with methicillin-susceptible *Staphylococcus aureus* (MSSA) and *Pseudomonas aeruginosa*. Inflammatory markers were normal, except for a slightly elevated ESR. Immunological evaluation showed reduced absolute CD8 + T cell and NK cell counts, low-to-normal B cell counts, and slightly low IgG and IgM levels. The CD4/CD8 ratio was also elevated. Management included IP (syrup, 3 mL three times daily for 14 days), inhaled gentamicin (7 days), inhaled beclomethasone dipropionate (ICS), inhaled fenoterol/ipratropium bromide, cetirizine, and iron supplementation. Clinical improvement was also observed.

### 3.4. Case 4

A 2.5-year-old girl with a history of congenital pulmonary airway malformation post-resection, recurrent bronchitis, and laryngitis developed acute laryngotracheobronchitis during hospitalisation. Polymerase chain reaction (PCR) was used to detect rhinovirus (RV) and human coronavirus (HCoV). The inflammatory markers were within normal ranges. Cellular immunity assessment demonstrated elevated absolute counts of T lymphocytes (CD3+, CD4+, and CD8+) and markedly elevated NK cell counts (absolute and percentage). The total IgG level was low. The CD4/CD8 ratio was within the normal range. The treatment involved IP (syrup, 3.5 mL four times daily for 14 days), inhaled beclomethasone dipropionate, salbutamol inhaled on demand, desloratadine, and a single dose of systemic dexamethasone. The patient exhibited a positive clinical response, with the duration of the respiratory illness being notably shorter than the typical clinical course for such a viral co-infection.

### 3.5. Case 5

A 9-year-old male with a complex primary immunodeficiency (ITK deficiency) following allogeneic haematopoietic stem cell transplantation (allo-HSCT), complicated by granulomatous lymphocytic interstitial lung disease (GLILD), bronchiectasis, chronic Epstein–Barr virus (EBV) infection (persisting significant PCR positivity in urine), and graft-versus-host disease (GvHD), developed acute non-obstructive bronchitis during a routine admission. PCR confirmed RSV infection. A post-discharge throat culture grew MRSA. Inflammatory markers (CRP levels) were normal. Immunological profiling showed elevated CD8+ T lymphocyte levels (absolute count and percentage), a low CD4/CD8 ratio, and significantly reduced natural killer (NK) cell numbers (absolute count and percentage). IP (500 mg three times daily for 14 days) was added to his chronic regimen, which included inhaled budesonide/formoterol (ICS/LABA; dose increased during RSV infection) and prophylactic trimethoprim/sulfamethoxazole (TMP/SMX). The patient remained clinically stable after the procedure.

### 3.6. Case 6

An 18-year-old male presented with severe bilateral cavitating pneumonia, right-sided pleural effusion (fluidothorax), and sepsis attributed to *Staphylococcus aureus*. Diagnostic investigations identified co-infection with influenza B virus and *Mycoplasma pneumoniae* (IgM and IgG positive). Inflammatory markers were markedly elevated initially but subsequently decreased. Cellular immunity assessment revealed lymphopenia, an elevated percentage of CD8+ T cells, and profoundly depleted NK cells (absolute count and percentage). The CD4/CD8 ratio was within the normal range. Intensive care management included intravenous antibiotics (cefotaxime and ciprofloxacin), initial intravenous dexamethasone, oxygen therapy, and immunomodulation with IP (1000 mg of granulate three times daily, administered for 20 days per month over 3 months). Gradual clinical improvement was observed, although the patient requested early discharge.

### 3.7. Case 7

An 8-month-old male infant with a history of prematurity, sepsis, left hydronephrosis, and anaemia presented with bronchial obstruction. During hospitalisation, he developed acute bronchiolitis, which culminated in partial respiratory insufficiency. PCR diagnostics confirmed a triple viral infection with RSV, PIV3, and adenovirus (ADV). Inflammatory markers (CRP) remained low despite the clinical severity. Immunophenotyping revealed NK cell depletion (absolute count and percentage) and an elevated percentage of CD8+ T lymphocytes, resulting in a low CD4/CD8 ratio. Management included IP (syrup, 2 mL, four times daily), intensified inhaled therapies (beclomethasone dipropionate and fenoterol/ipratropium bromide), oxygen support (high-flow followed by low-flow nasal cannula), a single dose of systemic methylprednisolone, and hepatoprotective agents. The patient was stabilised but required transfer for continued care due to slow recovery.

### 3.8. Case 8

A 14-year-old male with non-allergic asthma, a history of right lung lobectomy, and past (not current) immunodeficiency (hypo IgM, B-cell depletion) acquired SARS-CoV-2 infection during a routine hospitalisation. Inflammatory markers (CRP levels) were normal. Cellular immunity analysis indicated low absolute counts of CD3+ T cells, CD19+ B cells, and NK cells. The absolute CD8 + T cell count was low-normal, whereas the percentage was borderline high. The CD4/CD8 ratio was within the normal range. Management included IP (500 mg, four times daily for 14 days), continuation of inhaled fluticasone propionate, vitamin supplementation, and probiotics. The patient was discharged early for home isolation in a stable condition.

### 3.9. Case 9

A 1.5-year-old boy with a history of bronchial asthma and GERD presented for a control admission with acute bronchitis and pharyngitis. PCR confirmed RV and ADV co-infection, with serology positive for ADV IgM. Inflammatory markers (CRP and ESR) were elevated, accompanied by leukocytosis and lymphocytosis. Nasal culture yielded *Moraxella catarrhalis,* and throat culture yielded *Staphylococcus aureus*. Cellular and humoral immunity parameters were within normal limits, and specific IgE tests were negative. Treatment included IP (syrup, 3 mL three times daily for 14 days), inhaled fluticasone/salmeterol (ICS/LABA), cetirizine, and supportive care, including anti-reflux measures. The patient showed clinical improvement and was discharged early in a stable condition at parental request.

### 3.10. Case 10

An 11-month-old male with a complex history of surgically corrected oesophageal atresia with tracheoesophageal fistula, subsequent oesophageal stenosis requiring multiple dilatations, percutaneous endoscopic gastrostomy (PEG) tube placement, laryngomalacia, and recurrent bacterial pneumonia presented with acute bronchitis. PCR confirmed ADV infection, accompanied by positive ADV IgM serology results. Inflammatory markers (CRP and ESR) were significantly elevated initially but improved later. Marked reactive thrombocytosis was observed. Cellular immunity assessment revealed depleted absolute counts of CD3+, CD4+, CD8+ T lymphocytes, and NK cells, with an elevated CD4/CD8 ratio. Humoral immunity was also normal. Treatment involved IP (syrup, 2 mL three times daily for 14 days), transient systemic dexamethasone, inhaled fluticasone propionate, inhaled fenoterol/ipratropium bromide, fluconazole (for oral candidiasis), and intensive supportive care, including anti-reflux therapy (confirmed grade 4 GERD). Despite slow improvement and persistent cough, the patient was stabilised and discharged for continued care.

### 3.11. Case 11

A 2-year-old boy with recurrent obstructive bronchitis, tonsillitis, otitis media, atopic dermatitis, and confirmed aeroallergen sensitisation (dog, cat, guinea pig, and grasses) presented for evaluation. During hospitalisation, he developed acute lacunar tonsillitis and bronchitis associated with a PCR-confirmed triple viral infection: RV, ADV, and HBOV. ADV IgM was also positive. CRP levels were transiently elevated. Notable findings included marked peripheral blood eosinophilia and elevated serum total IgE levels. Cellular immunity showed a slightly low CD8+ T cell percentage but normal absolute counts and normal NK cells. Treatment comprised a prolonged course of IP (syrup, initially 3 mL four times daily, then reduced dosage for approximately two months total), cetirizine, acetylcysteine, anti-reflux therapy (magnesium alginate/simethicone for confirmed GERD), and emollients. The patient’s condition was stabilised.

### 3.12. Case 12

A nearly 3-year-old male with protracted cough, symptoms suggestive of GERD, cow’s milk protein IgE-mediated allergy, atopic dermatitis, and recurrent herpes labialis/stomatitis was admitted for diagnostic workup. During admission, the patient developed acute pharyngitis and herpetic gingivostomatitis. PCR confirmed RSV infection. Inflammatory markers (CRP and ESR) were normal. Cellular immunity revealed depleted B lymphocytes (low absolute count and percentage), a slightly low CD4+ T cell percentage, and elevated NK cells (absolute count and percentage). The humoral immunity parameters were normal. Nasal eosinophilia was also observed. Treatment included IP (syrup, 4 mL four times daily, duration determined by an immunologist), topical nasal antibiotics (neomycin/bacitracin), topical acyclovir cream, cetirizine, anti-reflux therapy (magnesium alginate/simethicone for confirmed extraesophageal reflux via salivary pepsin assay), and emollients. Rapid clinical improvement was observed.

### 3.13. Case 13

A 4-year-old boy with recurrent bronchitis, tonsillitis, otitis media, chronic sinusitis, epistaxis, suspected laryngopharyngeal reflux (EER), and possible selective IgA deficiency was evaluated. During hospitalisation, the patient developed acute lacunar tonsillitis and bronchitis. PCR detected RV and HCoV co-infection. The inflammatory markers were normal; however, eosinophilia was noted. Immunological assessment suggested IgA deficiency (low IgA level for age) with elevated IgG levels, and cellular immunity was largely normal. Nasal eosinophilia was also observed. Nasal cultures identified colonisation by *Moraxella catarrhalis* and penicillin-resistant *Streptococcus pneumoniae*. Treatment included a prolonged course of IP (syrup, initially 4.5 mL four times daily, then reduced dosage for two months total), topical ofloxacin drops, acetylcysteine, and supportive care, including anti-reflux measures and investigation for intermittent glycosuria. The patient was discharged in stable condition.

### 3.14. Case 14

A 5-year-old girl with recurrent laryngitis, allergic rhinitis, atopic dermatitis was admitted for evaluation. During her hospital stay, she developed acute laryngotracheitis. PCR confirmed RSV infection. CRP levels were borderline elevated acutely, with transient leukopenia and lymphopenia. Cellular immunity showed depleted absolute counts of CD3+ and CD8+ T lymphocytes, with a low CD8+ percentage. The CD4/CD8 ratio and NK cell counts were normal. Humoral immunity parameters were normal, and specific IgEs were negative, but nasal eosinophilia was significantly elevated (35%). Management included IP (syrup, initially 5 mL four times daily, then a reduced dose for 1 month total), inhaled beclometasone dipropionate, and acute symptomatic therapy (decongestants, antihistamines). Clinical stabilisation was achieved.

## 4. Discussion

The presented case series offers a detailed insight into the real-world application of IP as an adjunctive immunomodulatory therapy in a cohort of paediatric patients with complex, multifactorial ARIs. A primary observation from our analysis is the frequent presence of underlying cellular immune alterations, most notably the depletion of NK cells, which was identified in half of the cohort (Table 1). This immunological finding, coupled with the consistently favourable clinical course observed in all 14 patients following treatment regimens that included IP, forms the core of our discussion.

IP is a synthetic purine derivative, specifically formulated as a stable 1:3 molar complex combining inosine with pranobex (dimepranol p-acetamidobenzoate). Since its initial authorization in 1971, IP has achieved widespread use in numerous countries globally. The therapeutic effects of IP, including its antiviral activities, are predominantly attributed to its immunomodulatory properties (Figure 2), although the precise mechanisms remain multifaceted and not fully elucidated.

A salient feature across several cases in this series was the documentation of alterations in cellular immunity during the acute ARI episode. Depletion of NK cells, both in absolute counts and/or percentage, was a frequent finding, observed in half of the patients (7/14), who were infected with diverse viruses including RSV, PIV3, Influenza B, SARS-CoV-2, and ADV (cases 2, 3, 5, 6, 7, 8, 10). This aligns with the existing knowledge that viral infections can transiently impair NK cell function and numbers, potentially hindering early viral infection control [11,12]. This finding provides a strong immunological context for the therapeutic approach taken in this case series. Given that one of the most consistently reported effects of IP is the enhancement of NK cell activity, its use in patients with documented or suspected NK cell depletion is a targeted and rational therapeutic choice. This approach directly links the patient’s specific immunological alteration with the drug’s known mechanism of action [9,13].

Similarly, CD8+ T lymphocyte parameters varied considerably among the patients. Several cases demonstrated reduced absolute counts, while others showed elevated levels. CD8+ T cells are crucial for clearing virus-infected cells, but their dysregulation can contribute to immunopathology. IP’s reported ability to modulate T-lymphocyte cytotoxicity and restore T-cell function depressed during infection suggests a potential role in normalizing or supporting adaptive cellular responses in these scenarios [9,14,15]. Our observations are consistent with a broader body of evidence supporting the clinical and immunological efficacy of IP in paediatric respiratory infections. For instance, a study by Golebiowska-Wawrzyniak et al. on children with recurrent infections demonstrated that prophylactic treatment with IP led to a significant increase in the numbers of CD3+ and CD4+ T-lymphocytes and an improvement in their function, highlighting its restorative effect on cellular immunity [16]. Furthermore, reviews such as that by Abaturov et al. have summarized the evidence for IP’s dual mechanism, which involves both enhancing a Th1-mediated immune response and directly inhibiting viral replication, reinforcing the rationale for its use in children at risk of prolonged or complicated ARI courses [17]. These findings collectively support the immunomodulatory role of IP observed in our cohort, placing our results within the wider context of its established therapeutic potential.

It is imperative to address the question of causality. The retrospective and observational nature of this study, along with the absence of a control group, makes it impossible to establish a direct causal link between IP administration and the observed clinical improvements. The favourable outcomes could be attributed to the natural course of the illness, concomitant therapies, or other unmeasured factors. However, the goal of this report was not to prove superiority or efficacy, which would require a comparative cohort. Instead, the objective was to highlight the broad indicative base for IP, its favourable safety profile and tolerability, and to suggest that its targeted application in patients with suspected immune dysregulation may have potentially amplified the positive clinical course of their acute respiratory illnesses. The consistent pattern of clinical stabilisation in this high-risk, complex cohort is a noteworthy finding that warrants further investigation.

Notably, throughout the treatment periods described in this case series, IP therapy was well-tolerated by all patients. No adverse events directly attributable to IP were reported, confirming its favourable safety profile even in this complex, polymorbid cohort.

This study possesses several inherent limitations that must be acknowledged. Firstly, its retrospective case series design prevents the establishment of causality. Secondly, the significant heterogeneity within the patient cohort in terms of age, comorbidities, and viral pathogens limits generalizability. Thirdly, the lack of a contemporaneous control group makes it impossible to determine the relative contribution of IP to the observed outcomes. Finally, selection bias is present, as the decision to treat with IP was at the discretion of the attending physician, potentially favouring its use in more complex cases.

Despite these limitations, this case series provides valuable real-world evidence. It demonstrates that, in a specialised clinical setting, physicians are using prior immunological assessments to guide therapeutic decisions in complex ARI cases. The findings from this series generate a compelling hypothesis: targeted immunomodulation with IP, in paediatric patients with complicated ARIs and documented cellular immune alterations, may be a beneficial adjunctive strategy. Further prospective, randomized controlled trials are essential to definitively evaluate the efficacy of IP as an adjunctive therapy for specific types of viral ARIs in paediatric populations. Such studies should ideally stratify patients based on baseline immunological status and focus on elucidating the precise correlation between specific immunological parameters, such as NK cell function, and clinical outcomes.

## 5. Conclusions

In conclusion, this case series illustrates the use of IP in a diverse group of paediatric patients with complex viral ARIs, often characterised by underlying comorbidities and alterations in cellular immunity, particularly involving NK and CD8+ T cells. The selection of IP was guided by a strong immunological rationale based on its known mechanisms of action. While limitations preclude definitive conclusions, the findings support the immunological rationale for considering IP in selected cases and underscore the urgent need for well-designed prospective studies to clarify its role and optimise its application in this vulnerable population.

## Figures and Tables

**Figure 1 pediatrrep-17-00123-f001:**
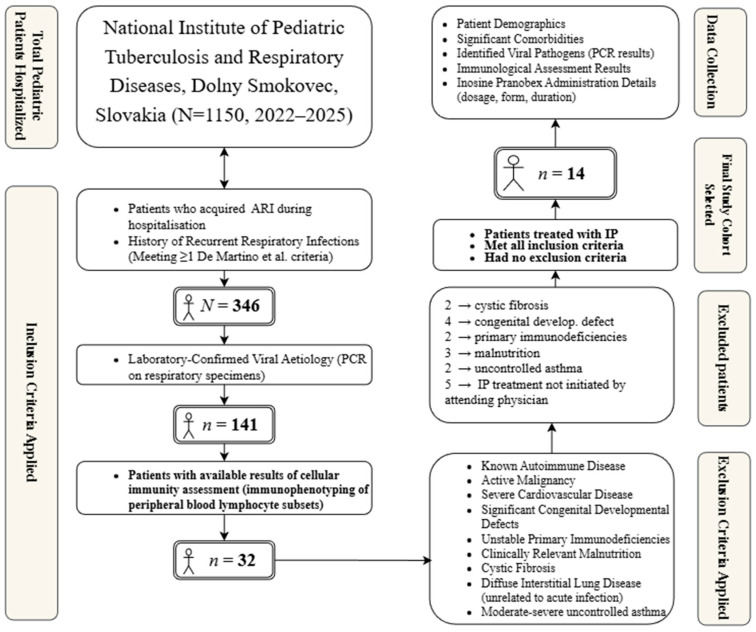
Flowchart diagram of clinical case series.

**Figure 2 pediatrrep-17-00123-f002:**
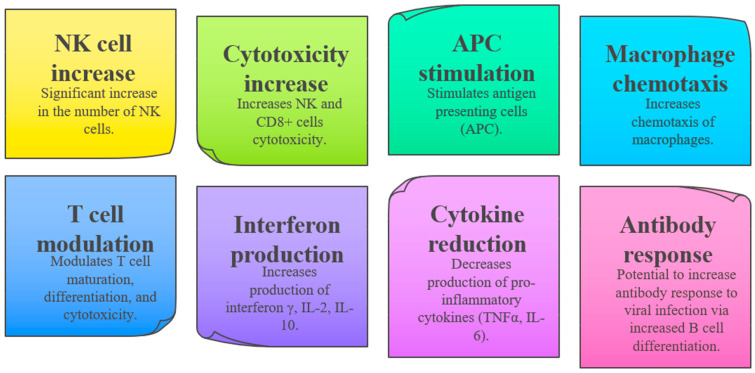
Inosine pranobex and immunomodulation properties. Abbreviations: **NK**: Natural Killer; CD8+: Cluster of Differentiation 8 positive; **APC**: Antigen Presenting Cells; **IFN-γ**: Interferon gamma; **IL-2**: Interleukin-2; **IL-10**: Interleukin-10; **TNFα**: Tumour Necrosis Factor alpha; **IL-6**: Interleukin-6.

**Table 1 pediatrrep-17-00123-t001:** Immune profile summary of clinical cases.

CASE	AGE	SEX	ACUTE DIAGNOSIS	DETECTED VIRAL PATHOGEN(S) (PCR)	CD4+ T CELLS (ABS/µL) (VS REF)	CD8+ T CELLS (ABS/µL) (VS REF)	NK CELLS (ABS/µL) (VS REF)	B CELLS (ABS/µL) (VS REF)	IRI (CD4/CD8) (VS REF)	IGG LEVEL (G/L) (VS REF)	IGA LEVEL (G/L) (VS REF)
1	17.5 mo	M	Obstructive Tracheo-bronchitis/ Incipient BPN	PIV3, HBOV	2270 (N)	1040 (N)	855 (N)	839 (D)	2.18 (Nl)	5.76 (N)	0.31 (N)
2	9 yr	F	Interstitial pneumonia	RSV Myc.	1301 (N)	590 (D)	67 (D)	818 (N)	2.20 (N)	12.50 (N)	1.72 (N)
3	7 mo	F	Acute obstructive bronchitis	PIV3	1955 (N)	593 (D)	204 (D)	643 (D)	3.30 (E)	3.27 (D)	0.15 (N)
4	2.5 yr	F	Acute laryngitis/ bronchitis	HCoV, RV	2602 (E)	1564 (E)	1918 (E)	1289 (N)	1.66 (N)	4.03 (D)	0.58 (N)
5	~9 yr	M	Acute non-obstructive bronchitis	RSV	1459 (N)	1594 (E)	162 (D)	939 (N)	0.92 (Low)	8.10 (N)	0.57 (N)
6	18 yr	M	Cavitating pneumonia	Influenza B (Myco+)	874 (N)	666 (N)	69 (D)	583 (E)	1.31 (N)	20.60 (E)	1.78 (N)
7	8 mo	M	Acute bronchiolitis	RSV, PIV3, ADV	1903 (N)	1767 (N)	195 (D)	915 (N)	1.08 (Low)	5.18 (N)	0.20 (N)
8	14 yr	M	COVID-19	SARS-CoV-2	830 (N)	499 (N)	281 (D)	290 (D)	1.66 (N)	8.85 (N)	0.82 (N)
9	1.5 yr	M	Acute bronchitis/ pharyngitis	RV, ADV	2122 (N)	2176 (E)	947 (N)	3233 (E)	0.97 (Low)	6.30 (N)	0.74 (N)
10	11 mo	M	Acute bronchitis	ADV	1138 (D)	613 (D)	241 (D)	1510 (N)	1.86 (N)	6.52 (N)	0.47 (N)
11	2 yr	M	Acute bronchitis/ tonsillitis	RV, ADV, HBOV	2395 (N)	1013 (N)	691 (N)	1680 (E)	2.36 (N)	6.85 (N)	0.29 (N)
12	~3 yr	M	Acute pharyngitis/ Herpetic gingivostomatitis	RSV	1140 (N)	1088 (N)	905 (E)	403 (D)	1.05 (N)	6.66 (N)	0.48 (N)
13	4 yr	M	Acute bronchitis/ tonsillitis	RV, HCoV	1053 (N)	847 (N)	693 (N)	1097 (N)	1.24 (N)	13.3 (E)	0.07 (D)
14	5 yr	F	Acute laryngo-tracheitis	RSV	721 (N)	556 (D)	503 (N)	582 (N)	1.30 (N)	9.03 (N)	0.77 (N)

Abbreviations: **ABS/µL:** Absolute count per microliter; **ADV:** Adenovirus; **B CELLS:** B Lymphocytes; **BPN:** Bronchopneumonia; **CD:** Cluster of Differentiation; **HBOV:** Human Bocavirus; **HCoV:** Human Coronavirus; **IgA:** Immunoglobulin A; **IgG:** Immunoglobulin G; **IRI:** Immunoregulatory Index (CD4/CD8 ratio); **mo:** Month(s); **Myco+:** Mycoplasma positive; **NK:** Natural Killer cells; **PCR:** Polymerase Chain Reaction; **PIV3:** Parainfluenza Virus type 3; **RSV:** Respiratory Syncytial Virus; **RV:** Rhinovirus; **SARS-CoV-2:** Virus causing COVID-19; **VS REF:** Versus Reference range; **yr:** Year(s); **(N):** Normal; **(E):** Elevated; **(D):** Depleted.

## Data Availability

Clinical data for the individual clinical cases are available from the authors upon request, subject to personal data protection requirements under the European Commission’s General Data Protection Regulation (GDPR).

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
