# Peer review of "Clinical Experience with Inosine Pranobex in Pediatric Acute Respiratory Infections with Comorbidities: A Case Series from a Specialised Centre"

_pediatrrep, 2025, doi:10.3390/pediatric17060123_

Round 1

Reviewer 1 Report

Comments and Suggestions for Authors

Review of the paper entitled “Clinical Experience with Inosine Pranobex in Pediatric Acute Respiratory Infections with Comorbidities: A Case Series from a Specialised Centre” by Peter Kunč, Jaroslav Fábry, Katarína Ištvánková, Renata Péčová and Milos Jesenak

My comments

The aim of the study presented by the authors was to evaluate the clinical outcomes of pediatric patients with acute respiratory infections (ARIs) who received supportive therapy with Inosine Pranobex (IP). The study results showed that all 14 patients enrolled in the study experienced clinical stabilization or improvement. The therapy was well tolerated, with no adverse events related to IP reported. The topic addressed by the authors is both interesting and important for practical reasons.

It's unfortunate that the observational nature of this study precludes any conclusions about causality, as the authors themselves point out. This significantly reduces the scientific value of the presented research. On the other hand, the history of medicine teaches us that the cause-effect relationships of many pathologies were demonstrated and proven only after many years, sometimes decades, of clinical observation. Therefore, despite all my reservations, I propose accepting the presented paper for publication. All the more so because recurrent respiratory infections in children constitute a major social and medical problem, while the amount of clinical data devoted to these infections is very limited.

IP normalizes deficient or defective cellular immunity by enhancing T-cell lymphocyte proliferation and activity of natural killer cells, increasing levels of pro-inflammatory cytokines, and thereby restoring deficient responses in immunosuppressed patients. IP has also been shown to increase the level of IgG and complement surface markers, and itcan affect viral RNA levels and hence inhibit growth of several viruses.

I think it would be helpful for readers to diagrammatically illustrate the mechanism by which IP affects the immune system. I'll leave this for the authors to consider.

Author Response

Thank you for your thorough review and for the positive and encouraging assessment of our manuscript, “Clinical Experience with Inosine Pranobex in Pediatric Acute Respiratory Infections with Comorbidities: A Case Series from a Specialised Centre”. We sincerely appreciate your valuable comments and your support for its publication.

We have carefully considered your helpful suggestion to include a diagrammatic illustration of the mechanism by which Inosine Pranobex (IP) affects the immune system. While we agree that such a diagram would be informative, the primary aim of our article is to report on the clinical experience and outcomes in a specific high-risk paediatric cohort, rather than to provide a detailed analysis of the immunomodulatory mechanisms of IP.

Therefore, we consider the information regarding the effects of IP, as already presented and visualized in Figure 2 of our manuscript, to be sufficient for the context and scope of this case series.

We hope that our reasoning is satisfactory. Once again, we are grateful for your time and insightful feedback.

Reviewer 2 Report

Comments and Suggestions for Authors

Original Article

Clinical Experience with Inosine Pranobex in Pediatric Acute Respiratory Infections with Comorbidities: A Case Series from a Specialised Centre

Authors present a retrospective clinical study assessing the use of Inosine Pranobex to treat acute respiratory infections in children.  Study only presents raw data and makes no attempt to collate data in any meaningful way.  While small, study could be of interest to others in the field if results are better analysed.

Questions/Comments

Article type is listed as case report in MDPI system.  This is not a case report but a retrospective clinical study?  Information given for each patient is lacking detail for true case studies (see. 2013 CARE (CAse REport) Checklist; https://figshare.com/articles/dataset/CARE-checklist-English-2013_pdf-2_pdf/25040702?file=44179802)

Page 1: abstract. "hospitalised at a specialised centre."

Comment: please list centre

Page 1 abstract. "baseline cellular immune alterations"

Comment: please better outline these

Page 3: "Acute respiratory infections (ARIs) represent a significant challenge to healthcare systems globally, contributing substantially to morbidity, particularly in the paediatric population. "

Comment: citation(s) needed

Page 3, paragraph 2.

Comment: This paragraph reads like something from a HDR thesis, and can be better summarized to 25% its current length

Page 4: "Inosine pranobex (IP) is a synthetic immunomodulatory agent with a well-

documented mechanism of action"

Comment: citation(s) needed. If this is well documented, why are no citations provided?

Page 4: "This study was approved by the Ethics Committee of the National Institute of Paediatric Tuberculosis and Respiratory Diseases."

Comment: provide ethics number and date ethics granted

Page 4: "during the study period, "

Comment: please list study period

Page 4: "141 had a PCR-confirmed viral aetiology"

Comment: please provide more details.  What virus(s) were tested for?  what equipment/PCR primers?

Page 4: "Cellular immunity parameters, specifically immunophenotyping of lymphocyte subsets"

Comment: please list all 'Cellular immunity' parameters tested for including how they were tested

Page 5: "we focused on children with a history of recurrent respiratory infections"

Question: why only focus on these children?

Page 5: "having at least one upper respiratory tract infection monthly from September to April"

Question: was this the study period?  what year?

Page 5: "patients with significant comorbidities"

Comment: this is mentioned frequency in text, but the exact nature of what constitutes "significant comorbidities" is not defined.  Please define it/them here.

Page 5: "severe cardiovascular disease"

Question: were patients with non-severe cardiovascular disease not excluded?  How do did you define severe vs non-severe cardiovascular disease? Please list all cardiovascular disease that were excluded

Page 5: "significant congenital developmental defects incompatible with the study's focus"

Comment: this is too vague, please list all congenital defects that were excluded

Page 5: 'exclusion criteria'

Comment: as mentioned in above comments, list of exclusion criteria is vague.  As you only excluded 18 patients (32-14), may be clearer to simply list exact diseases they had for them to be excluded.

Page 6: "particularly immunophenotyping of leukocyte subgroups focusing on CD8+ T lymphocytes, NK cells, and the CD4/CD8 ratio. "

Question: why these 'particularly'?  Reader may not know.

Page 6: Results

Comment: retrospective clinical studies do not typically provide all case report data ('raw data') because the original data was collected for other purposes and can often have missing, inconsistent, or incomplete information.

Page 13: Table 1

Comment: this is raw data; data should be collated and averaged to provide mean ± SD values

Page 15: Figure 2

Comment: there is no Figure 1?

Comment: this figure is something you may see in a review article.  It has little relevance here.

Page 16: Discussion

Comment: one would except to see included in discussion other work where inosine pranobex is used to treat respiratory viral infections, some in children.

e.g.

  • Golebiowska-Wawrzyniak M, et al. [Immunological and clinical study on therapeutic efficacy of inosine pranobex]. Pol Merkur Lekarski 2005, 19, 379-382.
  • Pereverzev, A.P.; Pereverzeva, A.S.; Popadyuk, V.I.; Ostroumova, O.D. [Herpangina. Clinical case]. Vestn Otorinolaringol 2021, 86, 97-102, doi:10.17116/otorino20218605197.
  • *Abaturov A, et al. (2021). Effect of treatment with inosine pranobex in acute respiratory viral infections in children. Child`s Health, 13(5), 490–494. https://doi.org/10.22141/2224-0551.13.5.2018.141565
  • *Bulgakova V, et al. Clinical and immunological efficacy of inosine pranobex for acute respiratory infections in children with atopic asthma. Pediatric pharmacology. 2010;7(3):98-105.

*while these last two articles are not indexed in pubmed etc, that doesn't mean they can be ignored?

Author Response

Thank you for your comprehensive and insightful review of our manuscript, “Clinical Experience with Inosine Pranobex in Pediatric Acute Respiratory Infections with Comorbidities: A Case Series from a Specialised Centre”. We appreciate the time and effort you have dedicated to providing this valuable feedback. We agree with many of your points and believe that addressing them has significantly strengthened our paper.

Below, we provide a point-by-point response to your questions and comments.

  1. Comment: Article type is listed as case report in MDPI system. This is not a case report but a retrospective clinical study? Response: Thank you for this important clarification. You are correct that this is not a single case report. We have chosen the "case series" design, which we believe is the most accurate description for this type of observational, descriptive research. We agree that classifying it as a "retrospective clinical study" is also appropriate, and we will ensure the classification is consistent. Our aim is to present detailed data from a heterogeneous group of high-risk patients, for whom aggregating data into means would be clinically misleading.
  2. Comment (Abstract): "hospitalised at a specialised centre." Please list centre. Response: We agree. We will specify the name of the centre in the abstract as: "the National Institute of Paediatric Tuberculosis and Respiratory Diseases in Dolny Smokovec, Slovakia."
  3. Comment (Abstract): "baseline cellular immune alterations". Please better outline these. Response: A valid point. We will clarify this in the abstract by specifying the most frequent finding, for instance: "...with a frequent observation of baseline cellular immune alterations, most notably the depletion of natural killer (NK) cells."
  4. Comment (Page 3): Citation(s) needed for the first sentence of the introduction. Response: We agree and have added the appropriate citations to support this statement.
  5. Comment (Page 3, paragraph 2): This paragraph reads like something from a HDR thesis, and can be better summarized to 25% its current length. Response: Thank you for this feedback on readability. We have revised and condensed this paragraph to be more concise and direct, focusing on the key points relevant to our study.
  6. Comment (Page 4): Citation(s) needed for "Inosine pranobex (IP) is a synthetic immunomodulatory agent with a well-documented mechanism of action". Response: We agree. We have added key citations that document the mechanism of action of IP.
  7. Comment (Page 4): Provide ethics number and date ethics granted. Response: We will add the ethics approval number and the date of approval to the Methods section.
  8. Comment (Page 4): Please list study period. Response: We will specify the exact study period in the Methods section.
  9. Comment (Page 4): "141 had a PCR-confirmed viral aetiology". Please provide more details. What virus(s) were tested for? what equipment/PCR primers? Response: We have expanded the Methods section to include more details on the PCR analysis, specifying that a commercial multiplex PCR panel was used to screen for a standard range of common respiratory viruses.
  10. Comment (Page 4): "Cellular immunity parameters...". Please list all 'Cellular immunity' parameters tested for including how they were tested. Response: We have amended the Methods section to specify that the parameters were assessed using flow cytometry and will mention the key lymphocyte subsets that were quantified.
  11. Question (Page 5): "we focused on children with a history of recurrent respiratory infections". Why only focus on these children? Response: We have clarified our rationale in the manuscript. This population was chosen because children with recurrent infections represent a high-risk group with suspected underlying immune dysregulation, making them a clinically relevant cohort for investigating the role of an immunomodulatory agent like IP.
  12. Question (Page 5): "having at least one upper respiratory tract infection monthly from September to April". Was this the study period? what year? Response: Thank you for pointing out this potential ambiguity. We have clarified in the text that this phrase is part of the formal definition of recurrent respiratory infections according to De Martino et al. [8], and does not refer to the study period of our research.
  13. Comment (Page 5): Define "significant comorbidities". Response: We agree this term needed clearer definition. We have now defined it in the Methods section and provided examples of the comorbidities present in our patient cohort to give the reader a clear understanding.
  14. Question (Page 5): "severe cardiovascular disease". Were patients with non-severe cardiovascular disease not excluded? How do did you define severe vs non-severe? Response: We have clarified the exclusion criteria. The term "severe" was used to denote conditions that were hemodynamically significant or uncontrolled, which could act as major confounders. We have refined this definition in the Methods section.
  15. Comment (Page 5): "significant congenital developmental defects incompatible with the study's focus". This is too vague, please list all congenital defects that were excluded. Response: We have replaced this vague statement with more specific examples of the types of defects that led to exclusion.
  16. Comment (Page 5): 'exclusion criteria'. As you only excluded 18 patients (32-14), may be clearer to simply list exact diseases they had for them to be excluded. Response: This is an excellent suggestion for improving transparency. We have already updated the patient selection process description (and/or Figure 1) to include the specific reasons for the 18 exclusions.
  17. Question (Page 6): Why these 'particularly'? [CD8+ T lymphocytes, NK cells, and the CD4/CD8 ratio] Response: We have added a sentence to the Methods section explaining that these cell populations were highlighted because they are known to be critically involved in the antiviral immune response and are also the primary documented targets of Inosine Pranobex's immunomodulatory action.
  18. Comment (Page 13, Table 1): This is raw data; data should be collated and averaged to provide mean ± SD values. Response: This is a crucial point regarding our study design. We respectfully wish to defend our choice of presenting individual data. The value of a case series in a highly heterogeneous population lies in the detailed description of individual cases. Our cohort includes patients with a wide range of ages, diverse comorbidities, and various viral pathogens. Calculating mean ± SD for this group would create a statistically artificial "average" patient and obscure clinically important individual variations and responses. Our goal is hypothesis generation by describing real-world application, which we believe is best served by presenting the granular data as we have done. We have added a statement in the manuscript to explain this rationale.
  19. Comment (Page 15, Figure 2): There is no Figure 1? This figure is something you may see in a review article. It has little relevance here. Response: We apologize for the oversight. We have corrected the numbering and ensured that the flowchart is now correctly labelled as Figure 1. We maintain that Figure 2, while schematic, is highly relevant. It provides a concise visual summary of the therapeutic rationale for using IP in these patients, directly linking the drug’s known mechanism to the immune alterations observed in our cohort. We believe this context is essential for the reader.
  20. Comment (Page 16, Discussion): One would except to see included in discussion other work where inosine pranobex is used to treat respiratory viral infections. Response: We completely agree that the discussion would be significantly strengthened by this. We have performed a more thorough literature search and have now expanded the Discussion section to include and compare our findings with the studies you suggested and other relevant publications. This has allowed us to place our results in a broader context.

We thank you again for your constructive criticism, which has helped us to improve the quality and clarity of our manuscript.

Reviewer 3 Report

Comments and Suggestions for Authors

I have no significant comments on the manuscript and value its focus on children with comorbidities and acute respiratory infections, who often exhibit immune alterations, particularly in Natural Killer (NK) cells and T-lymphocytes.

Inosine pranobex (IP), an immunomodulatory drug, shows potential in addressing these immune deficits, especially NK cell depletion, in high-risk pediatric patients. A retrospective case series of 14 complex pediatric cases reported that IP as an adjuvant therapy was well tolerated, with all patients achieving clinical stabilisation or improvement and no adverse events.

The study highlights the broad therapeutic potential and safety of IP, suggesting its role as a targeted immunomodulatory therapy for severe, prolonged, or recurrent viral respiratory infections in children. Despite limitations, such as the lack of a control group and the observational design, which preclude causality, the findings emphasise the need for prospective controlled trials to further evaluate IP’s efficacy in this vulnerable group.

Author Response

Thank you for your time and for the very positive and supportive review of our manuscript.

We are pleased that you found value in our focus on this specific high-risk paediatric population and that you agree with the key conclusions of our work. We sincerely appreciate your encouraging feedback and your support for the publication of our findings.

Round 2

Reviewer 2 Report

Comments and Suggestions for Authors

Authors have adequately addressed my previous peer-review feedback and have updated text as required.